# Strongly Improving the Sensitivity of Phosphorescence-Based Optical Oxygen Sensors by Exploiting Nano-Porous Substrates

**DOI:** 10.3390/bios12100774

**Published:** 2022-09-20

**Authors:** Chih-Yi Liu, Annada Sankar Sadhu, Riya Karmakar, Cheng-Shane Chu, Yi-Nan Lin, Shih-Hsin Chang, Goutam Kumar Dalapati, Sajal Biring

**Affiliations:** 1Organic Electronics Research Center, Ming Chi University of Technology, New Taipei City 24301, Taiwan; 2Department of Electronic Engineering, Ming Chi University of Technology, New Taipei City 24301, Taiwan; 3Department of Mechanical Engineering, Ming Chi University of Technology, New Taipei City 24301, Taiwan; 4MSSCORPS Co., Ltd., Hsinchu 300047, Taiwan; 5Sunkonnect Pte Ltd., 1 Cleantech Loop, Singapore 637141, Singapore

**Keywords:** anodic aluminum oxide, optical gas sensor, oxygen sensing, PtTFPP, PtOEP, sensitivity

## Abstract

Sensitivity is one of the crucial factors in determining the quality of a fluorescence/phosphorescence-based gas sensor, and is estimated from the measurement of responses (I_0_/I, where I_0_ and I refer to the measured optical intensity of a sensor in absence and presence of analyte molecules) at various concentrations of analytes. In this work, we demonstrate phosphorescence-based optical oxygen sensors fabricated on highly porous anodic aluminum oxide (AAO) membranes showing dramatically high response. These sensors exploit the enormous surface area of the AAO to facilitate the effective interaction between the sensing molecules and the analytes. We spin-coat an AAO membrane (200 nm pore diameter) with a platinum-based oxygen sensing porphyrin dye, platinum(II) meso-tetrakis (pentafluorophenyl) porphyrin (PtTFPP), to fabricate a sensor exhibiting I_0_/I ~400 at 100% oxygen atmosphere. To address the generality of the AAO membrane, we fabricate a separate sensor with another porphyrin dye, platinum octaethylporphyrin (PtOEP), which exhibits an even higher I_0_/I of ~500. Both of these sensors offer the highest responses as an optical oxygen sensor hitherto reported. SEM and EDS analysis are performed to realize the effect of the increased surface area of the AAO membrane on the enhanced sensitivity.

## 1. Introduction

Optical gas sensors are widely used in various sectors, including residential and industrial zones. These sensors are not only simple in design and achieve high sensitivity, but also offer a lot of analytical information by measuring various optical parameters via infrared and Raman spectroscopies (including surface-enhanced Raman spectroscopy), fluorescence, reflectometry, and absorptiometry methods [1]. Photoluminescence-based (fluorescence/phosphorescence) sensing have successfully drawn the attention of researchers because of its simple instrumentation, ease in sample fabrication, and straightforward measurement of optical signal variation caused by analytes. On the other hand, oxygen detection has been a primary research topic for a prolonged time because this gas is indispensable in various biochemical and chemical reactions [2]. By and large, all animals and plants generate energy by consuming oxygen present in ambient air [3,4]. Lower and higher concentration of oxygen in air lead to unconsciousness and even human deaths [5]. Oxygen toxicity occurs under prolonged exposure with oxygen partial pressure of ~0.5 bar or 50% oxygen at NTP (normal temperature and pressure) influencing cardiac, pulmonary, and neurologic systems of human beings [6]. Therefore, in the past several decades, quantitative analysis and monitoring of oxygen concentration is crucial in different areas such as environmental monitoring, chemical, and clinical fields [7,8,9]. In general, electrochemical sensors (both potentiometric and amperometric) including Clark-type sensors have been applied in the determination of oxygen concentrations [10]. The performance of conventional oxygen sensors has been improved further with the advances on solid electrolyte gas sensors and semiconductor gas sensors [11]. Nowadays, optical oxygen sensors have been frequently applied in environmental [12], clinical [13,14], and chemical fields [15,16,17].

Fluorescence/phosphorescence-based optical oxygen sensors monitor environmental O_2_ concentration by using the gas-induced quenching effect of the indicators [2], which are predominantly embedded in either a sol–gel [18,19,20] or polymer matrix [21,22,23]. Application of a sol-gel matrix has several advantages, such as good oxygen permeability, chemical and mechanical stability, and high optical transparency [24,25,26]. In addition, the porous structure of sol-gel matrix has even improved the sensitivity of platinum-(II)-complex-based oxygen indicators [18,19,20]. Two Pt-based porphyrin complexes, platinum tetrakis pentrafluorophenyl porphyrin (PtTFPP) and platinum octaethyl porphyrin (PtOEP), have been successfully applied in oxygen sensors because of their several salient features, such as large stokes’ shift (100–170 nm), high sensitivity, fast response, and good photochemical stability [18,19,20,21]. Many research works have been carried out recently to further increase the sensitivity of sensors by not only developing novel matrix materials, but by also incorporating nano/micro-particles in the matrix, or creating unique surface morphology of the substrate, to achieve a larger surface area for better interaction of gas molecules with the indicator molecules [27,28,29].

Recently, many gas sensors based on sensing films with a high surface area, which improves the interaction between indicators and analytes, have been reported. For example, the insertion of core-shell silica nanoparticles in PtTFPP film as an oxygen indicator achieved the optical response (I_0_/I, where I_0_ and I refer to the measured fluorescence/phosphorescence peak intensity of a sensor in the absence and presence of analyte molecules) of 166 at 100% O_2_ atmosphere [2]. Lee et al. improved photocurrent sensitivity by 61% through the morphological modification of the oxygen sensing PtOEP film embedding in polystyrene (pore diameter 300 nm to 1 µm) matrix, compared to solid film [30]. Recently, Mao et al. fabricated square arrays of PDMS-pillars as a porous substrate for a PtTFPP oxygen sensor to increase the surface area of interaction, achieving an optical response of 288 at 100% O_2_ and in the subsequent year, the optical response (~3) of PtTFPP oxygen sensor on flat solid films was improved to ~6.5 by introducing a honeycomb porous substrate [31,32]. The major research efforts to improve the response of an optical oxygen sensor are summarized chronologically in Table 1.

Porous anodic aluminium oxide (AAO) templates are composed of uniformly distributed nano-channels with hexagonally closed-pack arrangement (honey-comb structure), and thus provide large surface areas [43,44,45]. Furthermore, AAO substrate is highly reusable because of its strong resistant to heat and chemicals and extraordinary mechanical hardness [46]. Due to these unique advantages, an AAO template has a wide range of applications including gas sensing [47,48,49,50]. As per applications, AAO nano-channels can be filled with various materials for specific gas sensing [51,52,53,54,55,56]. Recently, Cui et al. developed a Ru (II) complex-based oxygen gas sensor using a commercially available AAO template showing the improved response of 10.47 at 80% O_2_ atmosphere [57]. In general, an AAO template fabricated by anodization of aluminium substrate possesses nano-channels with openings at one ends, while the other ends are closed by an oxide layer (called barrier layer). In an atmospheric environment, such channels are occupied by air, which opposes materials to fill in. Therefore, the desired materials are filled in by specific surface modifications [51,52]. This drawback can be overcome by using nano-channels with through-holes because filling material from one ends can easily push gas out from the other open ends.

In this study, we have used highly porous AAO through-hole membranes as substrates for oxygen sensing with PtTFPP dye molecules. The phosphorescence-based optical oxygen sensor was fabricated by spin-coating PtTFPP complex in an Octyl-triEOS/TEOS sol-gel matrix on the surface of AAO membrane. The extended surface area of the nano-channels facilitates interaction between gas and indicator molecules resulting in the dramatic improvement of sensitivity. The fabricated sensor exhibits optical response of ~400 in 100% oxygen environment, which is almost five times higher than that of a reference sensor (~80) based on a flat glass substrate. This is the highest response of an optical oxygen sensor hitherto achieved (Table 1). To realize the generality of through-hole AAO membranes as effective substrates for optical gas sensing, another phosphorescent molecules PtOEP were used to fabricate sensors which exhibit even higher optical response of ~500 in 100% oxygen environment. A detailed study on the performance of the two proposed sensors is presented in this article. This study demonstrates the potential of applying AAO through-hole membrane for an optical gas sensor to achieve higher sensitivity without compromising any critical parameters.

## 2. Experimental Procedure

### 2.1. Materials

We purchased through-hole AAO membranes (diameter of 13 mm and, pore size of 20 nm, 100 nm, and 200 nm) from Whatman (Kent, UK) and Sterlitech (Auburn, WA, USA), glass substrate (0.7 mm) from Corning (Corning, NY, USA), n-octyltriethoxysilane (Octyl-triEOS, 97.5%) from Alfa Aesar (Haverhill, MA, USA), Tetraethyl- orthosilane (TEOS, 99.5%) and Triton X-100 (analytical grade, 100%) from Acros organics (Geel, Belgium), tetrahydrofuran (THF, 99.9%) from TEDIA (Fairfield, OH, USA), EtOH (99.5%) from ECHO Chemical Co., Ltd. (Miaoli, Taiwan), HCl (32%) from Shimakyu (Taichung, Taiwan), Platinum(II) meso-tetrakis(pentafluorophenyl)porphyrin (PtTFPP) and platinum octaethylporphine (PtOEP) from Frontier Scientific (Logan, UT, USA), and Sigma-Aldrich (St. Louis, MO, USA), respectively. All the chemicals were used readily in the experiment without purification.

### 2.2. Preparation of Optical Oxygen Sensor

PtTFPP and PtOEP dye molecules were used separately to prepare the individual optical oxygen sensor on AAO and glass substrates. The oxygen sensing solution was prepared by dissolving 1 mg of dye powder (PtTFPP or PtOEP) in 1 mL of THF solution yielding a highly uniform dye solution [19,42]. The sol-gel matrix for the oxygen sensing molecules (PtTFPP or PtOEP) was prepared as follows. The precursor solution was prepared by mixing 4 mL TEOS and 0.4 mL Octyl-triEOS (C_8_-TEOS) solution together followed by adding EtOH (1.25 mL) and HCl (0.4 mL). This solution was then added to Triton X-100 (0.2 mL) and subsequently stirred magnetically for 1 h to achieve the homogeneity of the sol-gel [18,19]. Finally, 20 µL of the dye solution containing either PtTFPP or PtOEP and 10 µL of the sol-gel were mixed and then stirred vigorously for 10 min, resulting in oxygen sensing solution. The oxygen sensing solution was spin-coated on the surface of AAO membranes at 4000 rpm for 40 s. The dye molecule together with matrix material oozes out the through-hole nano-channels in the spin-coated process. Every sample was subsequently dried at room temperature for 24 h prior to use as an oxygen sensor, as schematically displayed in Figure 1. The reference sensor on glass substrate was fabricated by spin-coating a cleaned glass substrate (Corning) with the oxygen sensing solutions at 4000 rpm for 40 s and was dried at room temperature (27 °C) for 24 h following the similar process adopted in the fabrication of the AAO-based gas sensor.

### 2.3. Instrumentation

The experimental setup for the optical sensing measurements is illustrated in Figure 2. The optical image of the setup is provided in Appendix A. The excitation wavelengths of the dye molecules were determined by measuring the absorption spectra in UV-Visible spectrophotometer. The excitation light for the PtTFPP and PtOEP fluorophores was supplied from LED sources with peak wavelength of 405 and 380 nm, respectively. An arbitrary waveform generator (TGA1240, Thurlby Thandar Instruments (TTi) Ltd., Huntingdon, UK) at 10 kHz frequency was used to drive the sources. The emission spectra (fluorescence intensity) were captured by USB 4000 fiber optics spectrometer (Ocean Optics, Inc., Largo, FL, USA). The controlled oxygen environment was created by mixing O_2_ and N_2_ gases with mass flow controllers (Model GFC 17, Aalborg instruments and Controls Inc., Orangeburg, NY, USA) at room temperature (27 °C). A spin coater (SWIENCO, Taiwan) was used to prepare the thin films of the sensing materials on substrates. SEM (Phenom ProX, Thermo Fisher Scientific Inc., Waltham, MA, USA) and FESEM (JEOL, JSM-6701) were used to investigate the surface morphology and the elemental analysis via energy-dispersive X-ray spectroscopy (EDS).

### 2.4. Theory

The phosphorescence quenching of PtTFPP and PtOEP molecules by molecular oxygen occurs predominantly via Dexter energy transfer process [58]. The radiationless transition of the excited states of the fluorophores (PtTFPP/PtOEP) to its ground states in presence of oxygen molecules (quenchers) materializes through the transition of the ground state triplet O_2_ (^3^∑_g_^−^) to its excited singlet states (^1^∑_g_^+^ and ^1^Δ_g_). Ideally, the quenching behavior can be quantitatively described by the Stern-Volmer (S-V) equation [36], i.e.,
I_0_/I = 1 + K_SV_ [Q](1)

Here, I_0_ and I stand for the steady-state fluorescence intensities in the absence and presence of gaseous oxygen molecules; K_sv_ is the Stern–Volmer quenching constant; and [Q] is the concentration of oxygen molecules. It is obvious that I_0_/I versus [Q] i.e., S-V plot is linear with the slope of K_sv_. The linear S-V plot represents collisional quenching of the fluorophores in an ideal case. Nonlinearity of the S-V plots with both upward and downward curvatures is quite normal and have been observed in various environments [59]. Such curvatures originate in the simultaneous processes of dynamic and static quenching. In this scenario, the S-V equation is modified as [60]:I_0_/I = (1 + K_S_[Q]) (1 + K_D_[Q])(2)

Again, I_0_ and I stand for the steady-state fluorescence intensities in the absence and presence of oxygen molecules; K_D_ and K_S_ is the Stern–Volmer quenching constant for dynamic and static quenching, respectively; and [Q] is the concentration of oxygen molecules. In this work, Equations (1) and (2) have been applied to analyze our data.

## 3. Results

### 3.1. Characterization of AAO Membrane

The surface morphology of the AAO membrane were characterized by FESEM. The AAO surface shows randomly oriented nanopores with diameter of ~200 nm (Figure 3a). Note that the gaps between the pores are not well controlled and some of the pores are slightly overlapped. The cross-sectional image of the porous AAO membrane (Figure 3b) reveals the continuity of the long straight channels. The thickness of the AAO membrane is 60 µm (Figure 3c).

### 3.2. Optical Characterizations of the Sensing Material

The optical characterizations of the sensing dye molecules were performed by measuring the absorption and emission spectra of PtTFPP and PtOEP, as presented in Figure 4. The absorption spectrum of PtTFPP molecule shows a strong peak at 395 nm corresponding to Soret band, and two relatively short peaks at 505 nm and 540 nm corresponding to Q bands. Likewise, the absorption spectrum of PtOEP also exhibits one stronger peak at 380 nm belonging to Soret band, and two relatively shorter peaks at 508 nm and 540 nm belonging to Q bands. Therefore, LED light source with peak wavelength at 405 nm and 380 nm were selected to excite PtTFPP and PtOEP molecules, respectively. Upon optical excitation, the PtTFPP and PtOEP molecules emitted a strong phosphorescence with peak wavelength of ~650 nm and ~646 nm, respectively. A shoulder peak at ~720 nm was also observed in the PtTFPP spectrum [42].

### 3.3. Optical Characterizations of the Sensing Material

#### 3.3.1. O_2_ Sensing Properties of PtTFPP Oxygen Sensor

Two separate PtTFPP oxygen sensors were fabricated on different substrates i.e., glass and AAO membrane to realize the effect of increased surface area on the sensitivity by using excitation light source of 405 nm-LED. The phosphorescence peak of PtTFPP molecules at ~650 nm reduces dramatically in the presence of gaseous oxygen, as shown in Figure 5a. The change in phosphorescence intensity in the presence of O_2_ was recorded systematically at various concentrations of oxygen controlled by mixing N_2_ and O_2_ gases with proper ratio. Figure 5a indicates that the PtTFPP molecules on AAO membrane is strongly sensitive to O_2_ gas, since the phosphorescence intensity reduces severely even at a low concentration of 10%. The phosphorescence intensity decreases further with increasing O_2_ concentration, as shown by an enlarged plot in Figure 5b. Therefore, the PtTFPP/AAO substrate can be used as a powerful oxygen sensor. The phosphorescence intensity of PtTFPP on glass substrate also decrease with the increase in O_2_ concentration, but the decrease is much less compared to that on AAO substrate (Appendix A). For better comparison, we plotted the response (I_0_/I) vs oxygen concentration graphs for the glass and AAO cases, as shown in Figure 6. The peak intensities corresponding to I_0_ and I value were extracted by fitting spectra with multi-peak Gaussian curves to eliminate the contribution from the shoulder peak at ~720 nm (Appendix A) [42]. It is obvious that the response is linear for both the oxygen sensors (glass and AAO membrane), but the AAO based sensor exhibits much higher sensitivity compared to the glass-based sensor as evident from the difference in slopes of the data points. The data points are well-fitted with linear Stern-Volmer equation (Equation (1)), as shown by the solid lines in Figure 6. The linearity indicates that the sensing process predominantly follows static quenching mechanism. The K_sv_ value extracted from the fitting is 3.73%^−1^ for the AAO case, which is four times of that (0.86%^−1^) for the glass one. The enhanced response is attributed to the increased surface area of the AAO membrane compared to that of the glass substrate facilitating the interaction between analyte and dye molecules. As a consequence, the oxygen sensor based on AAO membrane exhibits response (I_0_/I) of ~400 at the oxygen concentration of 100%, which is almost 5 times higher compared to the one on glass substrate (~80). This is the highest response achieved by an optical oxygen sensor hitherto to the best of our knowledge (Table 1). It is worthwhile to mention that the 5 times increase in response is commensurate with the 5 times increase in surface area considering the effective length (nanochannels) of 1 µm (vide Appendix A).

#### 3.3.2. O_2_ Sensing Properties of PtOEP Oxygen Sensor

To realize the generality of the AAO-based optical gas sensor, we used another Pt-based porphyrin dye, PtOEP, embedded in sol-gel matrix to fabricate AAO-based sensor. Figure 7a shows the emission spectra of the PtOEP optical oxygen sensor on AAO membrane upon excitation by LED light with 380 nm wavelength at various oxygen concentrations. The intensity of the phosphorescent peak at ~646 nm reduces sharply with the increase in oxygen concentration from 0–100% indicating high sensitivity of PtOEP molecules towards gaseous oxygen. The zoomed-in spectra in Figure 7b clearly indicates that phosphorescence intensity drops severely with the increase of oxygen concentration and thus displays the efficiency of the sensor. For comparison, we also measured the phosphorescent spectra from another sensor of PtOEP molecules on glass substrate and observed a trend similar to that for the above-mentioned AAO case, as shown in Appendix A. However, the oxygen sensitivity for the glass-case was much less than that for the AAO one.

To quantitatively address the sensitivity of the proposed sensor, the response vs gas concentrations for the two cases was plotted as shown in Figure 8. The response of the AAO case shows linearly increasing trend at low oxygen concentrations and then increases rapidly at higher ones deviating from the linearity. Figure 8 also implies that AAO-based sensor is much more sensitive to oxygen concentration while comparing with the glass-based one. We fitted the data points with the non-linear Stern-Volmer equation (Equation (2)) as displayed by the solid lines in Figure 8. The good fittings of non-linear Stern-Volmer equation suggests that the static and dynamic quenching behaviors coexist in both sensors. The K_S_ and K_D_ values extracted from the fittings are 3.7%^−1^ and 0.003%^−1^ for the AAO case and 0.86%^−1^ and 0.002%^−1^ for the glass one, respectively. Note that the K_D_ values of both the AAO- and glass- based sensors are insignificant compared to the K_S_ values insinuating the predominance of static quenching over dynamic one in both cases. In addition, the K_S_ value of the AAO case is 4 times larger than that of the glass one, depicting the extraordinarily higher sensitivity of AAO-based sensors. Interestingly, the response of the AAO-based sensor (~500) at the oxygen concentration of 100% is nearly 5 times higher compared to that (~110) of the glass-based sensor. This is the highest response hitherto achieved by PtOEP based oxygen sensor to the best of our knowledge (Table 1).

### 3.4. Cross-Sectional and Elemental Analysis of AAO Nano-Channels

The extraordinarily high sensitivity of the AAO-based sensors requires an in-depth study to understand the mechanism behind it. To verify the presence of dye molecules in the nanochannels, we investigated the cross-sectional and elemental analysis of the AAO nanochannels via EDS. The cross-sectional SEM image of the AAO nanochannels filled with PtTFPP is presented in Figure 9a. It is hard to recognize PtTFPP layer in the SEM image. Therefore, we analyzed the sample by EDS. Figure 9b shows the EDS performed on the area marked by the red rectangle in Figure 9a. Although the EDS data clearly indicates the presence of Pt, F, and C elements from PtTFPP molecules along with the Al and O elements from the AAO membrane, but it could not distinguish the spatial distribution of the elements. To clarify this ambiguity, we carried out EDS mapping for all the individual elements in the area marked in the red rectangle in Figure 9a. Figure 9c displays the mapping which clearly indicates that all the elements are inside the AAO film and thus PtTFPP molecules are indeed inside nano-channels. Similar analysis for the AAO membranes coated with PtOEP dye molecules (Appendix A) also reveals the presence of PtOEP molecules inside AAO nano-channels unequivocally. The EDS measurements indicate the presence of dye (PtTFPP and PtOEP) molecules not only on the top surface of the AAO membrane but also inside the nano-channels. Such distribution of dye molecules provides more quencher-analyte interaction area in vertical direction when comparing with flat-substrate (e.g., glass) cases. In addition, dye molecules are highly accessible to the gas molecules entering from both ends of the through-hole membrane. More interactions cause higher responses leading to the extraordinarily high sensitivity for the AAO-based sensors.

### 3.5. Dynamic Response and Recovery of the Optical Oxygen Sensor

To investigate the performance of the optical oxygen sensors on AAO membrane for practical applications, we measured the response and recovery times of the proposed sensors. The dynamic responses of a PtTFPP and PtOEP- based oxygen sensor on AAO membrane were measured at different oxygen concentrations by switching between 100% N_2_ (0% O_2_) and systematically modified O_2_ environments as presented in Figure 10a and Figure 11a, respectively. Both graphs show the sharp fall and rise of the phosphorescence intensity upon exposure of oxygen gas and 100% N_2_ gas alternatively. Irrespective of the pre-treated O_2_ concentration, the restored intensities caused by 100% N_2_ processing is close to the initial value (intensity at 0 s) indicating strong recuperative nature of the sensors, which is favorable to reusability. The zoomed-in plots in Figure 10a and Figure 11a are used to quantitatively estimate the response and recovery time of the sensors, as shown in Figure 10b and Figure 11b. The estimated response times are 12 s and 10 s and the recovery times are 24 s and 46 s for the PtTFPP and PtOEP oxygen sensors, respectively. The response time is defined as the period required for intensity dropping down to the 90% of initial value, while the recovery time is estimated as the duration needed for intensity reaching the 90% of maximum value, as shown by the shaded columns in Figure 10b and Figure 11b. The fast response and recovery time (within several tenths of seconds) recommend the feasibility of the proposed oxygen sensors fabricated on AAO membrane in practical applications. Similar response and recovery measurements for glass-based sensors were also carried out, as shown in Appendix A. Roughly speaking, the response and recovery times for the two classes of substrates are varied by several tenths of percentage.

### 3.6. Photostability of Optical Oxygen Sensor

To shed some light on the lifetime of the proposed sensors, we measured the photostability of sensors under the exposure of continuous LED lights with the wavelengths of 405 nm for the PtTFPP devices and 380 nm for the PtOEP devices for 1 h at room temperature in ambient environment. The time-dependent intensity variations of LED light and phosphorescence of the AAO-based sensors are portrayed in Figure 12a,b, respectively. Both figures show stable intensities for long-term illumination. Careful observation reveals that the phosphorescence intensities of PtTFPP and PtOEP molecules decreased by 1% and 5%, respectively, after 1 h illumination. This phenomenon indicates that the sensor with PtTFPP is more stable than the one with PtOEP. The difference originates in the superior photostability of the PtTFPP molecules arising from the stability of the fluorophenyl groups attached to porphyrin compared to the relatively less stable ethyl groups attached to the porphyrin in PtOEP molecules [19].

In fact, the change in phosphorescence intensities observed in the photostability measurement is insignificant compared to the experimental errors and thus does not influence the accuracy in the sensing measurement. In addition, glass-based sensors also have reasonable photostability as demonstrated by similar measurements shown in Appendix A. Therefore, the photostability test confirms the optical robustness of the proposed oxygen sensors for successful practical applications.

Change in relative humidity and the ambient temperatures also affect the sensitivity of an optical gas sensor, which predominantly depend on the properties of indicator molecules. These issues have been studied rigorously on this system (PtTFPP and PtOEP sensor) showing insignificant change, with the change in relative humidity and reasonable changes with the change in ambient temperatures, as presented in our previous reports [61,62]. To test the humidity effect on our proposed sensor, we measured the responses at two different relative humidity—low (55%) and high (75%), revealing insignificant changes in responses under variations in humidity (Appendix A).

This study suggests that the increase in surface area is responsible for the increase in sensitivity. Therefore, it is intuitive to enlarge the surface area to its maximum limit for highest sensitivity. Since the surface area of the AAO nanochannels increases with the decrease in pore diameter, the smaller the pore diameter, the larger the surface area is. However, the dye molecules attached to the matrix have difficulty in entering into the narrower channels, limiting the sensitivity of the proposed sensor, as demonstrated in Appendix A. As a consequence, sensitivity on the AAO with a smaller pore diameter (20 nm) decreases and approaches that on the smooth glass substrate.

## 4. Conclusions

In this work, we have successfully demonstrated the high performance of optical oxygen sensors fabricated on readily available AAO membrane coated with dyes. The excellent performance originates from the large surface area of porous AAO membrane, which enhances interaction between oxygen and dye molecules. The response (I_0_/I) of the sensor based on PtTFPP dye on AAO membrane reaches ~400 at 100% O_2_ atmosphere, which is nearly 5 times higher than that (~80) on smooth glass substrate. In addition, the extraordinarily high response (~500) of PtOEP oxygen sensor on AAO membrane in 100% O_2_ atmosphere compared to that (~110) on glass substrate corroborates universal applicability of the AAO membrane in the design of optical gas sensors. These are the highest responses hitherto achieved by optical oxygen sensors. On the other hand, the response/recovery time for PtTFPP and PtOEP oxygen sensors on AAO membrane are 12 s/24 s and 10 s/46 s, respectively, which is fast enough for practical applications. In conclusion, this work presents a novel, reusable, cost-effective, and easy to fabricate optical oxygen sensor, with dramatically enhanced sensitivity by utilizing a unique porous substrate ingenuously.

## Figures and Tables

**Figure 1 biosensors-12-00774-f001:**
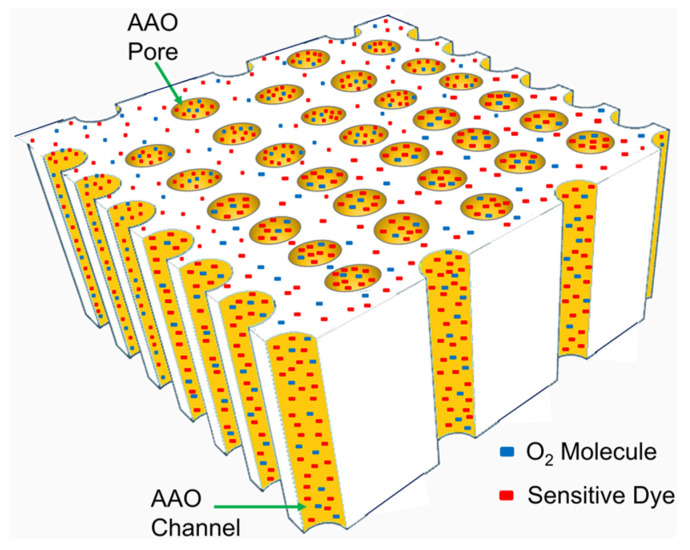
Schematic diagram of AAO through-hole membrane with dye molecules.

**Figure 2 biosensors-12-00774-f002:**
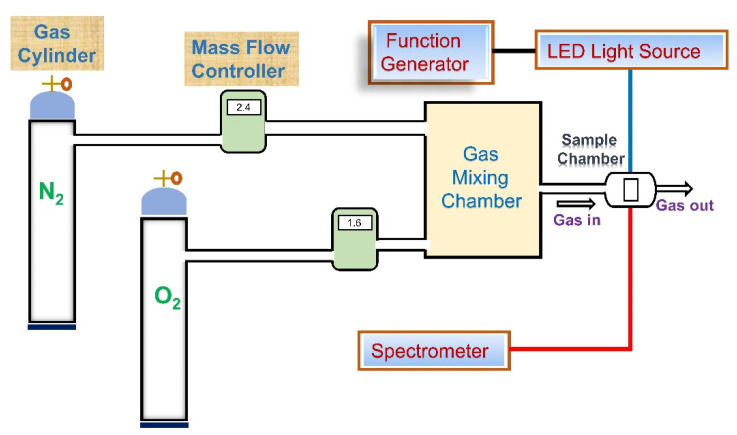
Schematic diagram of the setup for oxygen sensing measurements.

**Figure 3 biosensors-12-00774-f003:**
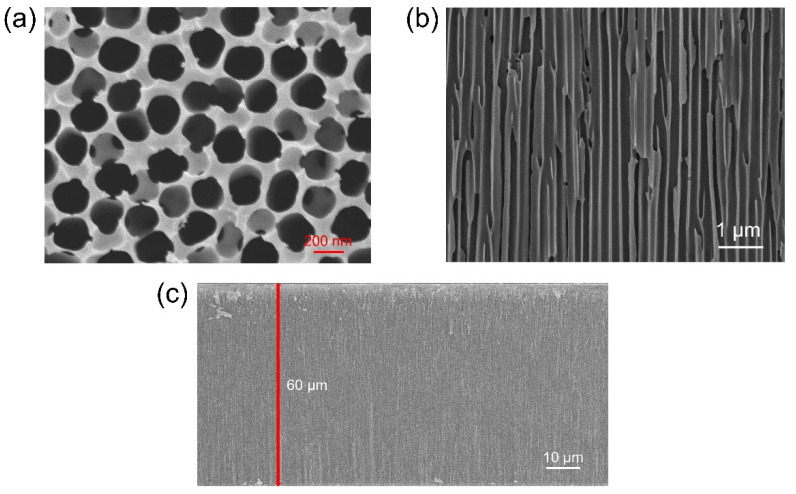
FESEM image of the AAO membrane from (**a**) topside and cross-sections with (**b**) small and (**c**) large scales. The membrane thickness is 60 μm as shown in (**c**).

**Figure 4 biosensors-12-00774-f004:**
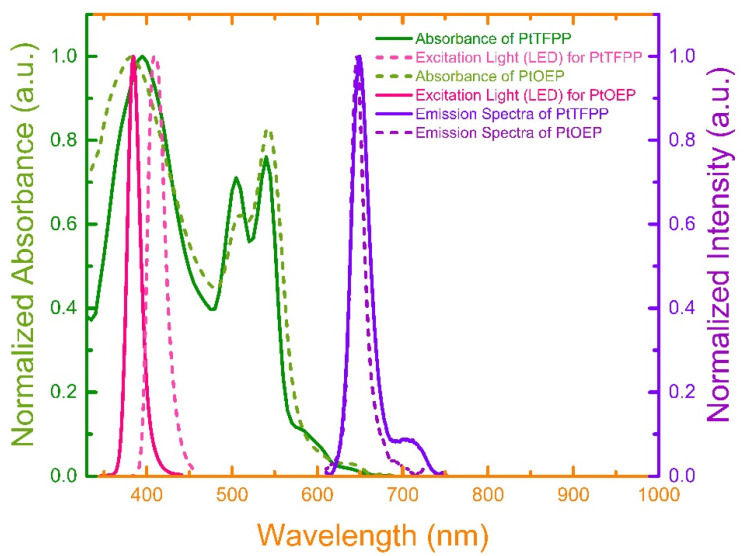
Absorption, emission, and excitation light (LED) spectra for PtTFPP and PtOEP dyes.

**Figure 5 biosensors-12-00774-f005:**
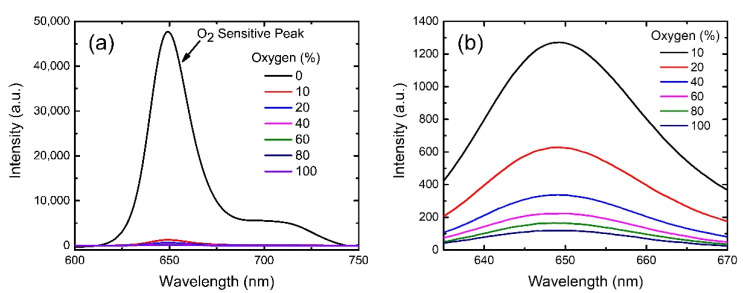
(**a**) Emission spectra of the PtTFPP molecules on an AAO membrane with increasing steps of oxygen concentrations from 0–100%. (**b**) Enlarged plot from (**a**) for better visualization.

**Figure 6 biosensors-12-00774-f006:**
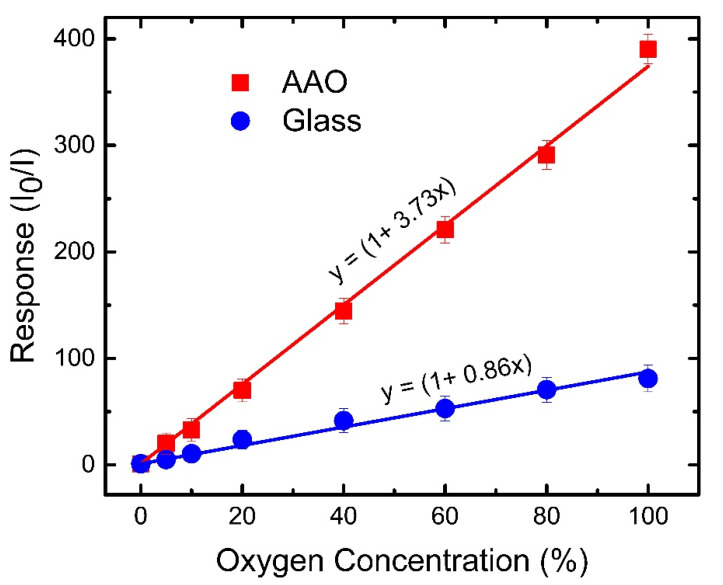
Responses as functions of oxygen concentrations for PtTFPP-based oxygen sensors by using AAO membranes (red squares) and glass substrates (blue dots). Equation (1) is used to fit the data points as shown by red (AAO case) and blue (glass case) lines. Y-axis error bars represent the standard deviations of the data from 10 samples.

**Figure 7 biosensors-12-00774-f007:**
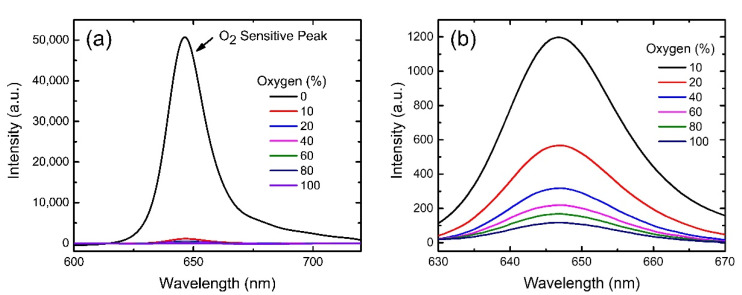
(**a**) Emission spectra of PtOEP molecules on AAO membrane with different oxygen concentrations from 0–100%. (**b**) Enlarged spectra from (**a**) for better visualization.

**Figure 8 biosensors-12-00774-f008:**
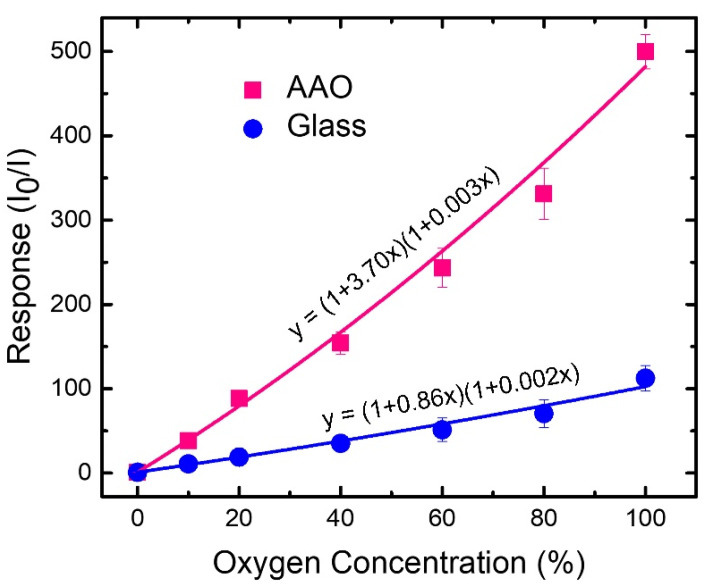
Responses as functions of oxygen concentrations for PtOEP-based oxygen sensors by using AAO membranes (pink squares) and glass substrates (blue dots). Equation (2) is used to fit the data points as shown by pink (AAO case) and blue (glass case) lines. Y-axis error bars represent the standard deviations of the data from 10 samples.

**Figure 9 biosensors-12-00774-f009:**
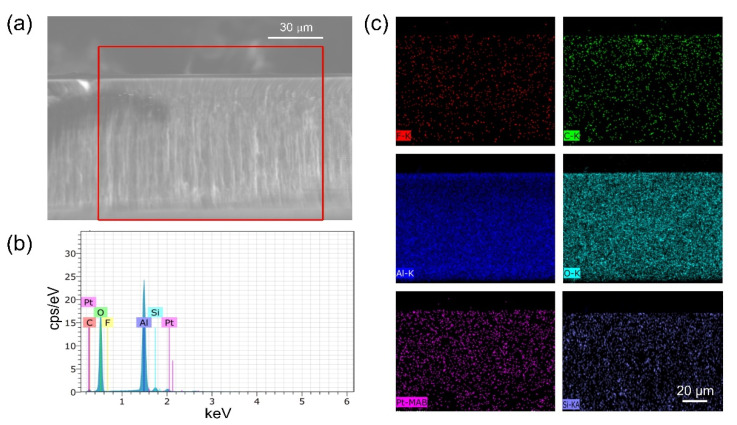
(**a**) Cross-sectional SEM image of the AAO nano-channels filled with PtTFPP molecule. (**b**) EDS spectra revealing Al, O, C, Pt, Si and F peak separately. (**c**) EDS mapping on the area marked by the red square in (**a**).

**Figure 10 biosensors-12-00774-f010:**
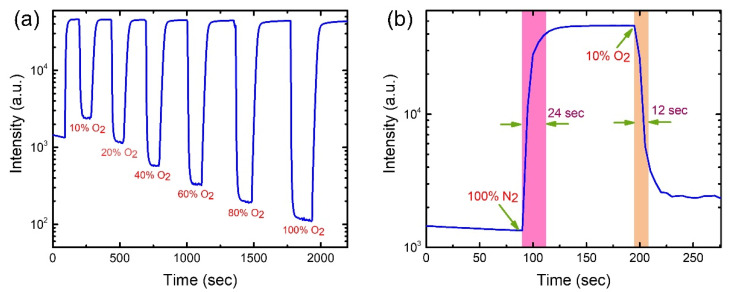
(**a**) Dynamic response of PtTFPP oxygen sensor on AAO membrane with gradual change in O_2_ concentrations. (**b**) A zoomed-in plot of (**a**) to estimate the response and recovery time.

**Figure 11 biosensors-12-00774-f011:**
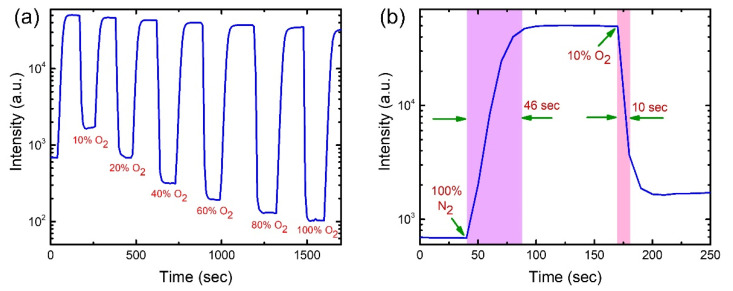
(**a**) Dynamic response of PtOEP oxygen sensor on AAO membrane with systematically modified O_2_ concentrations. (**b**) A zoomed-in plot of (**a**) to estimate the response and recovery time.

**Figure 12 biosensors-12-00774-f012:**
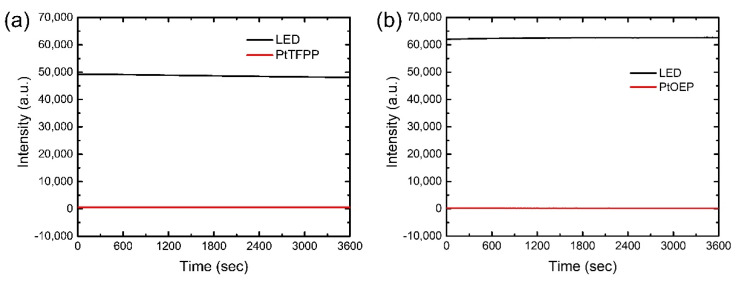
Photostability data of the oxygen sensors for (**a**) PtTFPP and (**b**) PtOEP on AAO membrane under 1 h continuous illumination.

**Table 1 biosensors-12-00774-t001:** Responses of various optical oxygen sensors are compared chronologically.

Substrate	Indicator/Matrix	Response (I0/I100)/ Characteristics	Year/Ref.
Optical fiber	PtOEP/n-propyl TriMOS/TFP-TriMOS	82.5/Linear SVPs	2007/[20]
Quartz glass	PtTFPP/Silica-gel	70/Linear SVPs	2010/[33]
Microscope glass	PtTFPP/Silica gel beads	1.4/Linear SVPs	2011/[34]
Optical fiber	[Ru(dpp)_3_]^2+^/Octyl-triEOS/TEOS and porous silica nanoparticles	26/Linear SVPs	2011/[35]
Optical fiber	PtTFPP/n-propyl-TriMOS/TEOS/Octyl-tri-EOS	155/Linear SVPs	2011/[36]
Optical fiber	PdTFPP/n-propyl-TriMOS/TEOS/Octyl-tri-EOS	263/Linear SVPs	2013/[37]
Optical fiber	PtTFPP/Octyl-triEOS/TEOS and metal-coated silica nanoparticles	167/Non-linear SVPs	2013/[38]
Optical fiber	PdTCPP/TEOS/Octyl-triEOS	153/Linear SVPs	2014/[39]
Aluminium foil or glass coverslips	PtTFPP/PSU-PCL	8.9/Linear SVPs	2015/[40]
Aluminium foil or glass coverslips	PtTFPP/PES-PCL	6.7/Linear SVPs	2015/[40]
Optical fiber	PdTFPP/TEOS/Octyl-triEOS and core–shell silica nanoparticles	386/Linear SVPs	2016/[41]
PDMS support layer	PtTFPP/PDMS pillar arrays	288/Linear SVPs from 0 to 1010 hPa O_2_	2017/[31]
Glass	PtTFPP/Ethyl cellulose	60 (Dual Sensor)/Non-linear SVPs	2019/[42]
AAO membrane	PtTFPP/TEOS/Octyl-triEOS	~400/Linear SVPs	This study
AAO membrane	PtOEP/TEOS/Octyl-triEOS	~500/Non-linear SVPs	This study

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
