# Peer review of "Strongly Improving the Sensitivity of Phosphorescence-Based Optical Oxygen Sensors by Exploiting Nano-Porous Substrates"

_biosensors, 2022, doi:10.3390/bios12100774_

Round 1

Reviewer 1 Report

The authors present an interesting study: combining specific dyes with porous anodic aluminum oxide (AAO) membranes to produce high performance of optical oxygen sensors. The logic of this work is novel. The research data is sufficient and the workload is relatively abundant. However before considering this work for publication, I have some comments or questions below:

1. In Introduction: Page 1 Line 37-40, “Photoluminescence-based (flu- orescence / phosphorescence) sensing have successfully drawn the attention of researchers because of its simple instrumentation, ease in sample fabrication, and straightforward measurement of optical signal variation caused by analytes.” Here, it is recommended to add relevant references. There are also many other sensors with porous materials and devices, for example, hydrogel optical devices [small 2015, 11, No. 24, 2869-2876], polymeric electrical sensors [ACS NANO. 2022, 16(1): 1511-1522], and even paper-based sensors [Light: Science & Applications (2016) 5, e16009]. A better introduction seems to need to prove the special merits of authors’ method by comparing with other scenarios.   

2. In Introduction: Page 2 Line 51-55, “The performance of the conventional oxygen sensors has been improved further with the advances on solid electrolyte gas sensors and semiconductor gas sensors [11]. Nowadays, optical oxygen sensors have been frequently applied in environmental [12], clinical [13,14] and chemical fields [15–17].” The description of the sensor is confusing and does not make a connection with the importance of oxygen concentration monitoring mentioned in the previous sentence. Without a clear logic, different types of sensors are simply piled together.

3. For Table 1, I suggest to add the legend.

4. Does the regularity of the nanopore also affect the monitoring performance of the sensor? The nanopore regularity shown in the experimental part of the manuscript is very low and not as periodic as expected in Scheme 1.

5. Page 7 Line 237-239, “The enhanced re- 237 sponse is attributed to the increased surface area of the AAO membrane compared to that 238 of the glass substrate facilitating the interaction between analyte and dye molecules.” Relevant references are suggested here.

6. Page 8 Line 255-256, “To realize the generality of the AAO-based optical gas sensor, we used another Pt-based porphyrin dye, PtOEP, embedded in sol-gel matrix to fabricate AAO-based sensor.” The reasons for choosing these 2 materials in the manuscript can be properly explained (ie, why they can demonstrate the generality of the AAO-based optical gas sensor).

7. Page 10 Line 303-306, “To clarify this ambiguity, we carried out EDS mapping for all the individual elements in the area marked in red rectangle in Figure 9a. Figure 9c displays the mapping which clearly indicates that all the elements are inside the AAO film and thus PtTFPP molecules are indeed inside nano-channels.” How did the author perform precise mapping of individual elements, using point scanning? If a spot sweep is used, how can it be determined that the spot is on the nanochannel at this magnification?

8. Why are the scales of the SEM and EDS images inconsistent in Figure 9?

9. Format of reference 58 is not consistent.

Reviewer 2 Report

I read this manuscript with interest. The following comments may help the authors to improve the manuscript before acceptance.

1.       Figure 1, what is the thickness of the membrane?

2.       Would it be possible to include a digital picture (photo) of the experimental setup?

3.       It would be helpful for the readers if the authors could explain the mechanism of how this photoluminescence works.

4.       In the introduction part, it is useful if the authors can compare the advantage of phosphorescence to fluorescence. The following relevant references can be added for more information about the literature on the use of fluorescence in sensing:

Golabi, Mohsen, et al. "Development of reverse transcription loop-mediated isothermal amplification assay for rapid and on-site detection of avian influenza virus." Frontiers in Cellular and Infection Microbiology 11 (2021): 652048.

Round 2

Reviewer 2 Report

I do not have other comments.